

# Evaluation of unsupervised static topic models' emergence detection ability

Xue Li[1], Ciro D. Esposito[2], Paul Groth[1], Jonathan Sitruk[2],
Balazs Szatmari[2] and Nachoem Wijnberg[2,3]

[1] Informatics Institute, University of Amsterdam, Amsterdam, Netherlands
[2] Business School, University of Amsterdam, Amsterdam, Netherlands
[3] University of Johannesburg, Johannesburg, South Africa

## ABSTRACT

Detecting emerging topics is crucial for understanding research trends, technological advancements, and shifts in public discourse. While unsupervised topic modeling techniques such as Latent Dirichlet allocation (LDA), BERTopic, and CoWords clustering are widely used for topic extraction, their ability to retrospectively detect emerging topics without relying on ground truth labels has not been systematically compared. This gap largely stems from the lack of a dedicated evaluation metric for measuring emergence detection. In this study, we introduce a quantitative evaluation metric to assess the effectiveness of topic models in detecting emerging topics. We evaluate three topic modeling approaches using both qualitative analysis and our proposed emergence detection metric. Our results indicate that, qualitatively, CoWords identifies emerging topics earlier than LDA and BERTopics. Quantitatively, our evaluation metric demonstrates that LDA achieves an average F1 score of 80.6% in emergence detection, outperforming BERTopic by 24.0%. These findings highlight the strengths and limitations of different topic models for emergence detection, while our proposed metric provides a robust framework for future benchmarking in this area.

## INTRODUCTION

Topic extraction, also known as topic modeling, is the process of automatically identifying and extracting meaningful latent concepts or topics from a collection of documents or texts (*Blei, 2012*). It is a fundamental technique in natural language processing. Unsupervised topic extraction is commonly used because it does not rely on predefined categories or labeled data.

Topic emergence refers to the appearance of new groups of words representing a topic within textual data. New topics frequently emerge as fields and communities change and it is useful to be able to detect these new topics. For example, science progresses on the principle that new and emerging topics overshadow the older ones (*Kuhn, 1962*). In a social context, organizations are keen to be updated with the new trends that could affect their operations, internally through email (*Wang & McCallum, 2006*) and externally using social media platforms (*Chen et al., 2013*). Therefore, understanding which topic model is capable of better detecting topic emergence is beneficial for many applications in academia, entrepreneurship, and government.

Corresponding author
Xue Li, x.li3@uva.nl

Recent advances in topic modeling have focused on developing models that extract topics that are more coherent, diverse, and interpretable (*Abdelrazek et al., 2022*; *de Melo & Merialdo, 2024*). However, there is limited research comparing the ability of these models to detect emerging topics. In other words, when extracting topics and analyzing their evolution over time, we aim to select a model that can identify emerging topics earlier than others. Specifically, we focus on an unsupervised setting where the ground truth of the topics is unknown.

While previous studies have explored forecasting topic emergence during the "embryo" stage using revolutionary networks and citation-based models (*Salatino, Osborne & Motta, 2018*), others have analyzed retrospective trend detection by identifying shifts in topic proportions or leveraging change-point detection techniques (*Boutaleb, Picault & Grosjean, 2024*; *Rahimi et al., 2023*). However, these models they primarily introduce specific models for topic emergence detection without systematically comparing how well different unsupervised static topic models, such as Latent Dirichlet allocation (LDA), BERTopic, and CoWords, detect topic emergence. To address this gap, we evaluate the ability of multiple topic models to detect emerging topics in a unsupervised setting systematically. We propose a generic framework that can be applied to any topic models, and an independent evaluation metric to assess their effectiveness.

Comparing topic models presents several challenges. Ideally, if different topic models were to extract the same topics, we could compare the trends produced by each model and determine which model captures the emergence of a topic earlier by having human evaluators evaluate the associated trends. While this approach can be effective, there are still limitations:

(1) Different topic models possess different assumptions and initialization procedures, which may lead to different levels of granularity in the learned topics. As a result, there might not be a one-to-one correspondence between topics across different models.
(2) The lack of ground truth makes it difficult to apply a systematic comparison approach.

To address these limitations, we propose two evaluation methods. The first involves human evaluation of top-matched topics across models, which provides a qualitative analysis of the generated topics and when they emerge. The second is a quantitative, independent measure of a model's ability to detect emerging topics without ground truth. This measure leverages the generative nature of each model and uses the predictions of emerging topics from a global model as a silver standard for comparison. The measure uses the global model as a benchmark that captures the overall representations of topics, allowing us to assess whether local models accurately reflect the emergence of topics when trained on a subset of data.

To this end, our pipeline consists of four main steps: topic extraction, topic matching, qualitative analysis, and independent emergence performance measure. In the topic extraction step, we input a set of time stamped documents and a static topic model, producing a list of topics with their evolution over time and a trained model as outputs. For topic matching, we use two methods to indicate the prevalence between any pair of topics:

Top Words Overlap Rate, based on extracted top words, and the Kullback-Leibler (KL) divergence, based on topic-document probability distribution. In the qualitative analysis, we inspect the extracted top words. Lastly, we develop an emergence performance measure that can be applied independently across static topic models. This measure uses a global model trained on the entire dataset as a silver standard and compares its predictions with those from local models trained on time-based snapshots of the data. While each generative model can produce topics and assess whether they are emerging, aligning topics from different models is challenging. To overcome this bottleneck, we use documents as a proxy and calculate the agreement between the global and local models on whether a document is emerging.

In this work, we analyze three classic static topic models, which are widely adopted in real-world applications: CoWords clustering (*Callon, Courtial & Laville, 1991*; *Callon et al., 1983*), Latent Dirichlet Allocation (LDA) (*Blei, Ng & Jordan, 2003*) and BERTopic (*Grootendorst, 2022*). They also represent three prominent categories of topic models: algebraic, probabilistic, and neural (*Abdelrazek et al., 2022*). CoWords clustering utilizes word co-occurrence patterns, LDA leverages probabilistic modeling, and BERTopic employs neural embeddings to capture rich topic representations. We compare extracted topics across different models. We also exhibit and compare different matching strategies across different models and present topic emergence among different methods.

To perform these comparisons, we conduct comprehensive experiments on three datasets: Web of Science bio-medical publications, ACL anthology publications (*Rohatgi, 2022*), and the Enron email dataset (*Klimt & Yang, 2004*). These datasets cover diverse domains and contain varying degrees of topics. Examining models on these datasets allows us to compare their effectiveness and generalizability across contexts with varying levels of topic institutionalization.

The development of measuring and comparing the topic model's ability to topic emergence in the natural language processing (NLP) community can be of help to different communities, including firms (company-wide emails), governments and scientists (scientific abstracts) (*Kwon et al., 2019*), in choosing the model for detecting changing topics. Additionally, this framework can be applied to patent analysis and other types of text analysis (*Mckinnon & Rubino, 2022*). Another application area is innovation policy, where the identification of emerging/changing scientific topics is of great interest to governmental policymakers, which can be used to foster national-level scientific competitiveness (*Schot & Steinmueller, 2023*).

Summarizing, the main contributions of our work are as follows:

- *A systematic framework for topic emergence detection.* We introduce a comprehensive pipeline that extracts topic, match topics across different models and track them for their emergence. Unlike previous studies that focus on a single topic modeling, our framework enables direct comparison of multiple models in an unsupervised setting.
- *The first comparative study for static topic models for detecting emerging topics.* We perform a qualitative analysis comparing how three widely used topic models capture

emerging topics. This is the first study that systematically benchmarks different static topic modeling approaches for emergence detection.

- *A novel evaluation metric for topic emergence detection.* We propose an unsupervised metric that assesses a model's ability to detect emerging topics without the need for manual assessment. This metric enables scalable evaluation across diverse textual domains.

## RELATED WORK

With respect to topic extraction, prior work has typically relied on CoWords clustering (*Bai, Li & Liu, 2021*; *Zhang et al., 2022*; *Choudhury & Uddin, 2016*) and probabilistic topic models like LDA (*Li, Chen & Wang, 2021*; *Chen et al., 2017*; *Qi et al., 2018*; *Blei & Lafferty, 2006*) to extract topics from textual data (*e.g.*, scientific publications, emails) and track their evolution through time. Recent advances, particularly the development of pre-trained large language models like BERT (*Devlin et al., 2018*; *Grootendorst, 2022*; *Dieng, Ruiz & Blei, 2020*), have paved the way for new and promising applications in topic extraction.

Evaluating topic models presents a multifaceted challenge that hinges on both the accuracy and the relevance of the topics generated. Traditional metrics such as perplexity and coherence scores have been foundational, providing quantitative benchmarks that assess the internal consistency and semantic similarity within topics (*Röder, Both & Hinneburg, 2015*; *Newman et al., 2010*). However, these metrics often fall short in capturing the practical utility of the topics in real-world applications. More recent approaches emphasize user-centric evaluations, where the interpretability and applicability of topics to specific tasks are assessed through user studies or expert validations (*Chang et al., 2009*; *Lau, Newman & Baldwin, 2014*). Such methodologies aim to bridge the gap between statistical performance and practical significance, ensuring that the topics are not only coherent but also meaningful and actionable in specific contexts. Despite these advances, there is very limited development in evaluating topic model's ability to detect topic emergence.

Within the literature on topic modeling, there is a line of work that focuses on dynamic topic modeling (DTM) (*Blei & Lafferty, 2006*), which explicitly incorporates the time associated with the documents in the modeling process. The topic representation at timestamp $t$ depends on the representation at $t - 1$. DTM focuses on how topic representations evolve over time, offering different representations given time slices. Although DTM is capable of modeling how the concept of a topic shifts, it does not assist in detecting topic emergence. The downstream applications of such models often focus on tracking the dynamics or development of topics (*Leskovec, Backstrom & Kleinberg, 2009*; *Greene & Cross, 2017*) instead of detecting emergence. In contrast to our work, we use a retrospective approach focused on emergence that can learn a global representation of the topic and compare it with local representations, using a document-level proxy for emergence detection.

Recent work on emerging topic detection has explored both forecasting-based and retrospective approaches. Forecasting-based methods aim to predict the rise of emerging topics before they become widely recognized, leveraging historical trends and external

signals. Augur (*Salatino, Osborne & Motta, 2018*) introduced a forecasting-based approach leveraging evolutionary networks to predict topic emergence before their recognition. Other forecasting methods utilize incremental topic modeling (*Gerasimenko et al., 2023*) and anomaly detection in topic distributions (*Redyuk, 2018*) to detect early signals of emerging research trends. These approaches typically rely on time-series modeling, citation networks, or machine learning-based trend extrapolation to anticipate future developments.

On the other hand, retrospective approaches focus on identifying when topics have already emerged in historical data. Methods such as BERTrend (*Boutaleb, Picault & Grosjean, 2024*) and ATEM (*Rahimi et al., 2023*) apply neural topic modeling and graph-based embeddings to detect topic shifts in past corpora. Additional studies explore word embedding trajectory monitoring (*Christophe et al., 2021*) and structural changepoint analysis (*Bose & Mukherjee, 2021*; *Wang & Goutte, 2018*) to identify when new topics emerge based on shifts in semantic space or statistical deviations in topic prevalence.

However, these methods typically introduce a single model without benchmarking multiple topic modeling techniques, making it difficult to compare the relative effectiveness of different approaches. Furthermore, other works on topic model evaluation primarily focus on coherence and perplexity (*Kherwa & Bansal, 2021*) or even topic coverage (*Korencic et al., 2020*), yet these metrics fail to capture a model's ability to detect emerging topics. In contrast, our work systematically compares three widely used static topic models (LDA, BERTopic, and CoWords) for emergence detection across multiple domains and introduces an independent evaluation metric that does not rely on external metadata or predefined ground truth labels.

Similar to our work, novelty detection also uses documents as a proxy for change. Novelty detection aims to find text with new information compared to what has been seen or known before (*Ghosal et al., 2022*). Prior studies frame novelty detection as a document-level binary classification problem, where documents are classified as "novel" or "not novel" based on a set of existing documents (*Ghosal et al., 2021*; *Saikh et al., 2017*; *Nandi & Basak, 2020*). Our work is different in that we first identify emerging topics using a trained model subsequently, we determine if a document is emerging if it is associated with an emerging topic. This approach differs from assessing if topics are evolving based on a single model (*Uban, Caragea & Dinu, 2021*).

## TOPIC EXTRACTION PIPELINE

To study method performance on emergence detection, we develop a pipeline that takes a dataset with time as input and then outputs matching topics and their change over time. Our general framework is shown in Fig. 1. The pipeline consists of these steps: data preprocessing, topic modeling, topic and trends extraction, topic matching, and finally, matched topics emergence analysis.

The goal of the pipeline is to generate and compare topics with their trends using varying static topic models in an unsupervised manner. The final output will be topics with their associated trends that are generated and matched using different topic models. Once

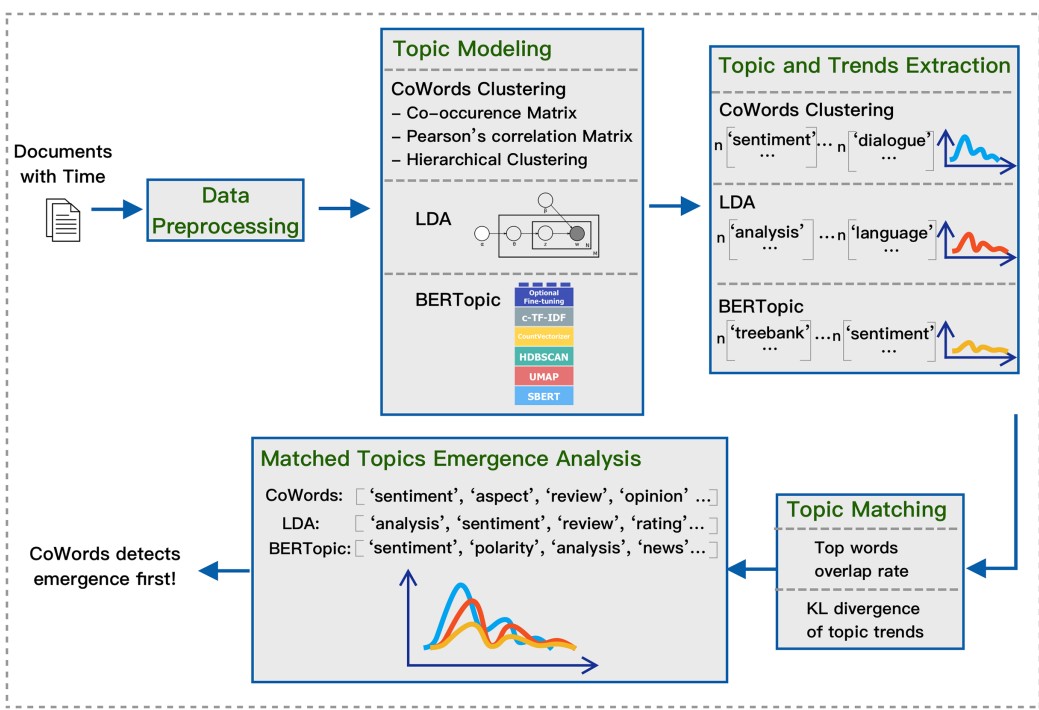

**Figure 1 Pipeline for extracting and matching topics with three models.**

the matched topics and their trends are generated, we can then apply qualitative analysis and quantitative analysis on which model is better at emergence detection.

## Overview of the pipeline

The input of the pipeline is a set of documents with a time stamp for each document. First, we apply the preprocessing procedure for each dataset following natural language processing practices. We then train each model on the processed texts to generate topics. For each topic model, we fix the number of topics for each dataset to 100. Any semantic space can be divided into an unlimited number of latent spaces. A topic can also be divided into different latent concepts. We fix the number of topics to attempt to ensure the statistical and semantic meaningfulness of the topics and a balanced trade-off between topic variation and interpretability. We use the number 100 because it is aligned with previous study (*Uban, Caragea & Dinu, 2021*).

After the models are trained with 100 topics, we subsequently extract 20 top words for each topic as the representation of that topic. The top words are selected based on frequency to ensure they are diverse and representative. We then apply a hierarchical topic-matching strategy based on two metrics: top words overlap rate and KL divergence over the matched trends. After the topics are matched, we plot the trends and then apply both qualitative evaluation and quantitative evaluation on topic emergence detection ability. The code and data used in this article are available at: https://zenodo.org/records/14503316. We now explain each of the steps in details in the following sections.

## Data preprocessing

To reduce the noise coming from uninformative words, the raw text of the merged titles and abstracts has been pre-processed. Pre-processing can drastically impact extraction quality. Hence, we describe our entire preprocessing process in the following seven steps with detailed parameter specifications.

**Step 1.** The date/year of the document, the title of the document and abstracts/email bodies of the document are needed for further analysis. Therefore, we remove entries where any of these fields are empty. Additionally, for computational reasons, we remove documents where the length is more than 700 tokens for the Enron dataset.

**Step 2.** Each document is tokenized and converted into a list of lowercase tokens. Tokens shorter than two characters and longer than 15 are discarded in the tokenization process. Accents and punctuation are removed.

**Step 3.** Documents are further de-noised by removing English stopwords, *i.e.*, frequently used words that do not provide significant distributional information.

**Step 4.** The vocabulary is enriched with bigrams, *i.e.*, pairs of consecutive tokens that often appear together (*e.g.*, "amino acid", "frontal cortex", *etc.*). We use pointwise mutual information (PMI) as a score function and a threshold of 100, and the minimum collective frequency for a valid bigram is 20.

**Step 5.** We perform lemmatization to group the inflected forms of a word in a single token.

**Step 6.** After performing part-of-speech tagging, we keep only nouns, adjectives and verbs.

**Step 7.** We removed from the remaining tokens all the words that occur in more than 25% of the documents or in less than 0.01% of the documents.

## Topic modeling

We select three models for comparison: (1) CoWords clustering, (2) LDA, and (3) BERTopic. We follow the categorizations from *Abdelrazek et al. (2022)*, selecting the most widely used topic model from the three prominent categories: algebraic, probabilistic, and neural. Comparing these three models provides us a intuition on how the three types of topic models perform on topic emergence detection.

### CoWords clustering

The idea behind CoWords clustering is that the co-occurrence of words describes the contents of the texts (*Callon, Courtial & Laville, 1991*). Based on this notion, methods have clustered words in the keywords lists, titles and abstracts, or other publication data fields, using multivariate statistical techniques, such as factor analysis, principal component analysis, and hierarchical clustering to obtain topics (*Wang et al., 2012*). In this work, we utilize hierarchical clustering for topic modeling.

### LDA

As a probabilistic topic model, the basic assumption of LDA is that the words are generated according to a mixture model where the mixture proportions are random, and the mixture components or topics are shared by all documents (*Blei & Lafferty, 2006*). It is based on the

idea that documents contain multiple topics, intended as distributions over a fixed vocabulary (*Blei, 2012*, p. 78). However, one limitation of LDA is that it models natural language as bag-of-words, discarding the word orders in the document.

### BERTopic

As one of the most widely used neural topic models, BERTopic is popular for its adaptability. It utilizes the recent development in NLP, modeling sentences with word orders using sentence embedding models such as sentence-BERT. The semantic space created by sentence-BERT can be seen as a continuous space of sub-topics. We can discretize this space by detecting high-density areas and associating them with a topic. To do so, we use dimensionality reduction such as Uniform Manifold Approximation and Projection (UMAP) and clustering techniques such as Hierarchical Density-Based Spatial Clustering of Applications with Noise (HDBSCAN). UMAP is a manifold learning technique that is good at preserving both global structure and local structures. HDBSCAN (*Campello, Moulavi & Sander, 2013*) is a density-based, hierarchical clustering method that is noise-aware (*i.e.*, potentially outliers are not forced to belong to a cluster and could be labeled as noise) and based on soft assignment (*i.e.*, each point is associated to its cluster with a confidence score). Moreover, HDBSCAN relaxes the need to set the number of clusters as a hyperparameter, requiring specifying only the minimum number of clusters desired. Once the clusters have been identified, we can retrieve topic embeddings by calculating the mean of all the documents belonging to the same cluster, *i.e.*, the centroids. Since we learn word embeddings and document embeddings jointly in the same semantic space, we can look at the K words closer to the centroid to get a set of representative terms for that topic. This process is called fine-tuning the topic representation. Specifically, in our work, we use the Maximal Marginal Relevance algorithm to reduce the redundancy of the extracted keywords in each topic.

After extracting topics with each method, we observed that some topics have few documents associated with and some topics have only digits as top words. We filter out these outliers.

## Topic and trends extraction

Once the model is trained, topics can be extracted using different representations. In this work, we represent each topic by its top words, providing an interpretable summary of the topic's content. Additionally, we track the evolution of each topic over time by quantifying the number of documents associated with it at different time points. This allows us to analyze topic trends, identifying patterns of emergence and decline across the dataset. However, the absolute number of documents associated with a topic may not be a reliable indicator of its popularity, as the total number of documents per year varies. To address this, we use prevalence as a normalized measure of topic popularity. Prevalence quantifies topic prominence by dividing the number of documents assigned to a topic by the total number of documents in that year, ensuring a fair comparison across different time periods. This normalization allows us to accurately compare topic trends over time, independent of fluctuations in document volume.

## Topic matching

Once the topics and their associated trends are extracted, comparing topics across different models requires a robust matching strategy. Since topic models may learn different representations of the same underlying concepts, we need to align topics from different models to conduct a fair evaluation of their ability to detect emerging topics.

We employ two matching strategies in this work:

- **Top Words Overlap Rate (TWOR):** for matching topics based on their most representative words.
- **Kullback–Leibler (KL) divergence:** for matching topics based on their temporal trends.

The intuition behind TWOR is that topics are primarily characterized by their most representative words, making an exact match based on word overlap a straightforward way to determine topic similarity. Meanwhile, KL divergence quantifies the similarity of topic prevalence distributions over time, under the assumption that similar topics exhibit similar temporal trends.

### Top words overlap rate

The first matching strategy is based on comparing the top words of each topic. For each extracted topic, we retrieve the top $n$ words that are most representative of that topic. These words are selected based on Term Frequency-Inverse Document Frequency (TF-IDF) scores, which measure a word's relevance in the *corpus*.

In BERTopic, a cluster-level TF-IDF variant (c-TF-IDF) is used, calculating the weights as term importance at a cluster level:

$$\text{ctfidf}(t, c) = \text{tf}(t, c) \cdot \log\left(1 + \frac{A}{f_c}\right), \tag{1}$$

in which $t$ is the term, $c$ is the cluster, $tf(t, c)$ is the term frequency of term $t$ within cluster $c$, indicating how often term $t$ appears within that cluster. $A$ is the average number of words per cluster.

After obtaining the $n$ top words for each topic $T$, we calculate the overlap rate between any two topics:

$$\text{TWOR}(i, j) = \frac{T_i \cap T_j}{N}, \tag{2}$$

where $N$ is the number of words for each topic.

### KL divergence

The second score function is Kullback–Leibler divergence for topic trends. This function is selected under the assumption that similar topics will have similar trends over time. We calculate topic prevalence based on the number of documents associated with the topic given any timestamp:

$$\text{Prevalence}(t_i, y_j) = \frac{N_{ij}}{N_j}, \tag{3}$$

in which $N$ is the number of documents given topic $i$ and year $j$. We then apply KL divergence between two distributions:

$$D_{KL}(P||Q) = \sum_i P(i) log \frac{P(i)}{Q(i)} \tag{4}$$

### Hierarchical topic matching strategy

Because topic models differ in how they define topics, there is no direct one-to-one correspondence between topics across models. Instead, each topic from one model can have partial overlap with multiple topics from another model, creating an N-to-N mapping problem. To address this, we use a hierarchical topic matching approach:

- First we match BERTopic topics ($T_{bertopic}$) to LDA topics ($T_{lda}$) using one of the above matching methods (TWOR or KL divergence).
- Next, we match CoWords topics ($T_c w$) to the already aligned BERTopic-LDA topics ($T_{lda\_bertopic}$), instead of directly matching all three models at once.
- To resolve conflicts in the N-to-N mapping, we prioritize matches by selecting topics that maximize the sum of matching scores, ensuring that the most representative topics are aligned across models:

$$max \sum (S_{lda\_cw}, S_{lda\_bertopic}). \tag{5}$$

Through this hierarchical strategy, we reduce complexity and maintain coherence in topic alignment, making it possible to systematically compare the models' ability to detect emerging topics.

## Matched topics emergence analysis

We employ two analysis for evaluating models' emergence detection ability: qualitative approach and quantitative approach.

### Qualitative comparison

Although all these methods have achieved outstanding performance on a variety of datasets, comparing them directly on the matched topics is still challenging. When the one-to-one mapping of topics exists across models, then we can directly compare their evolutions and derive the ability for early emergence detection. However, different topic models exhibit different assumptions and initializations, often resulting in the extraction of topics with different levels of granularity. For example, in the NLP context, one topic extraction model might extract topics with top words such as *['dependency', 'output', 'tree', …]* and *['parse', 'syntactic', 'class', 'contextual', 'syntax', …]* which corresponds to two different NLP tasks "dependency parsing" and "syntactic parsing", a different model might extract only one topic with top words *['structure', 'parse', 'syntactic', 'dependency', …]*, which represents the parsing task in general. Therefore, we perform a topic matching with two distinct score functions, Top Words Overlap Rate and KL-divergence, and then evaluate the top-matched topics qualitatively.

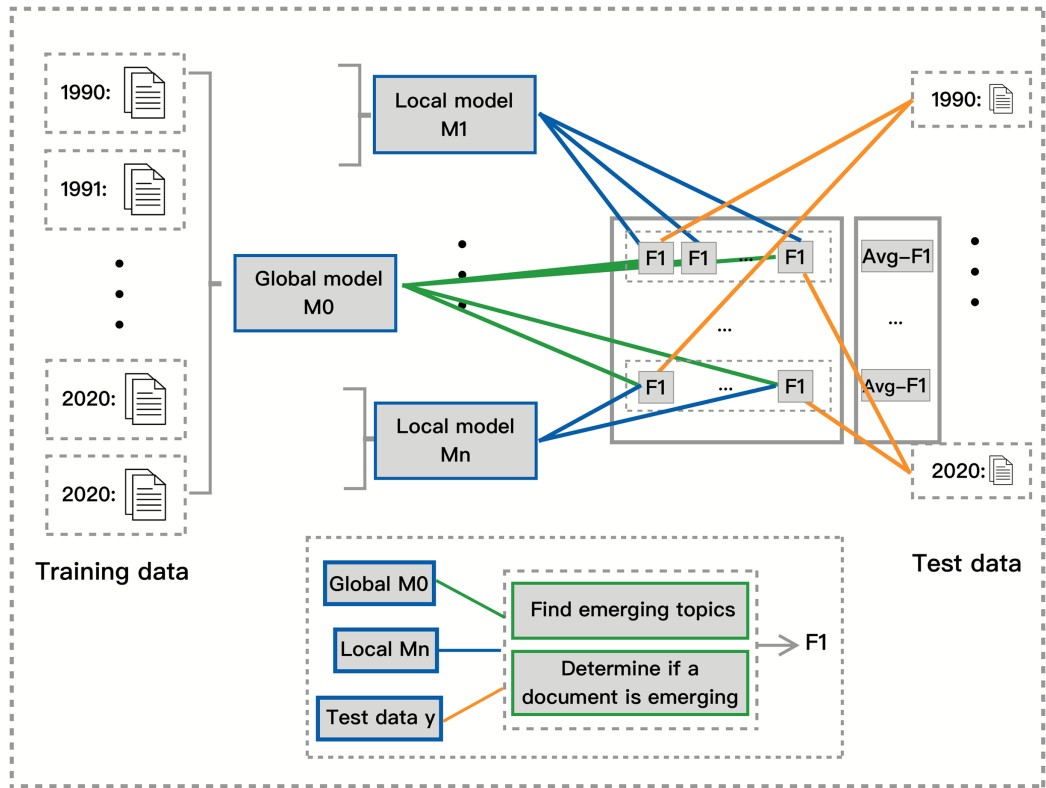

**Figure 2** Independent quantitative measure between global and local model.

### Quantitative evaluation metric

In this section, we describe our proposed approach for measuring a model's ability to detect emerging topics without relying on ground truth labels. Unlike traditional topic modeling evaluations that focus on coherence scores or perplexity, our approach quantifies how well a model captures topic emergence over time by assessing the agreement between local and global models. This method is summarized in Fig. 2.

Detecting emerging topics is challenging due to the lack of predefined ground truth labels that indicate when a topic becomes significant. Existing evaluation metrics such as topic coherence measure the semantic quality of topics based on word co-occurrence patterns, while perplexity evaluates how well a model predicts unseen text. However, these metrics do not assess whether a model effectively identifies when a topic is emerging. Our approach addresses this limitation by introducing a framework that compares local and global topic models to infer a model's emergence detection ability.

To evaluate a model's ability to detect emerging topics, we train a global model on the full dataset and local models on segmented time slices and then compare their predictions. The global model serves as a silver standard, enabling us to assess how well each local model captures emerging topics within its respective time period. As illustrated in Fig. 2, the orange lines represent test data from a given time span, the green lines indicate

emerging documents classified by the global model, and the blue lines indicate emerging documents classified by the local models.

We begin by segmenting the dataset based on timestamps and splitting each segment into training (80%) and test (20%) data. We then train a global topic model on the entire dataset to provide a comprehensive reference for topic emergence. Simultaneously, we train local topic models on each time segment's training data and evaluate them on test data from all other segments. We compute precision, recall, and F1 scores to measure the agreement between local and global models, providing a quantitative assessment of how well each model detects topic emergence.

Since there is no predefined ground truth for emerging topics, we introduce the concept of emerging documents. Rather than directly tracking topic-level trends, we classify a document as emerging when its dominant topics are identified as emerging by the global model. This enables a systematic evaluation of how well local models align with the global model in identifying emerging topics at a document level.

A higher F1 score between local and global models indicates greater agreement, meaning that the local model successfully captures emerging topics in a manner similar to the global model. Higher F1 scores also suggest that the model is more sensitive to topic changes and better at tracking emerging topics within shorter time spans. In contrast, lower F1 scores indicate that the local model struggles to capture emerging trends visible in the full dataset, implying that it is less responsive to short-term topic evolution. Models with lower F1 scores may be better suited for capturing long-term topic distributions rather than detecting short-term emergence.

Using document-level agreement instead of topic-level comparisons allows us to capture real-world knowledge shifts, making it a meaningful proxy for topic emergence. Since topic models assign probability distributions to documents, tracking the emergence of topics through documents provides a granular assessment of how topics gain prominence over time. Additionally, this method avoids the limitations of direct topic-level comparisons, where different models may produce topics at varying levels of granularity.

While this approach provides a robust framework for retrospective emergence detection, it has some limitations. The effectiveness of the metric depends on the granularity of learned topics—if topics are too broad, emerging documents may be harder to identify. Additionally, the global model's predictions may introduce bias, as it could overfit dominant research trends and fail to recognize smaller, emerging topics. Another limitation is the lack of external validation; while we use the global model as a silver standard, future work could incorporate expert annotations or alternative reference points, such as external event timelines.

Overall, our quantitative evaluation metric provides an objective and scalable method for assessing topic emergence detection. By leveraging document-level agreement between local and global models, we introduce a ground-truth free approach that is applicable across different datasets and topic models.

### Finding emerging topics

Given a topic trend, determining if a topic is emerging in a certain time period fundamentally relies on if the trend shows an upward trajectory. By definition, a topic emerges when it gains prominence over time, in which the most direct way to quantify it is by computing the growth rate. This aligns with other retrospective methods of identifying emerging topics (*Gerasimenko et al., 2023*). While other forecasting-based methods utilizes external information such as citation graphs, their emerging detection methods are not directly applicable in our setting.

In this work, we calculate the growth rate between any given timestamp $t_i$ and $t_j$; if the growth rate is positive, the topic is emerging; otherwise, it is not emerging. For topic $k$, we determine if $k$ is emerging in the time period of $i$ and $j$, we use:

$$g_k(t_i, t_j) = \frac{C_{ik} - C_{jk}}{C_{ik}}, \tag{6}$$

in which $C_{ik}$ is the number of documents associated with topic $k$ at time $t_i$, reflecting how frequently the topic appears in that period. Topic $k$ is emerging is $g_k(t_i, t_j) > 0$, $k$ is not emerging otherwise.

Specifically, for each given time period, we first find the emerging topics. Then, we associate each document with topics based on the predicted probability of the model. If a document is associated with topics that are determined to be emerging by the model, it is an emerging document. We calculate the F1 score based on the agreements between the global and local models on the same set of documents.

## EXPERIMENTAL SETUP

We detail our experimental settings here. We apply our pipeline to three datasets from different domains, using three topic models from different categories. We then match topics and evolutions extracted from three models, and subsequently analysis their topic emergence detection.

### Datasets

We used three datasets to validate our experiments: Biomedical publications on the Web of Science (WoS), anthology publications of ACL (ACL), and the Enron email dataset. These three datasets cover the biomedical, natural language processing, and corporate communication domains. Both WoS and ACL are scientific publications, presenting more structured language usages, while Enron dataset represent more fuzzy language usage. Additionally, WoS and ACL datasets represent different domains, showing how domains could potentially affect topic extraction. For each dataset, we preprocess the raw text following the steps in Figure, and the statistics of each dataset after pre-processing are shown in Table 1.

### Web of science

The WoS dataset contains 171,499 publications from the *Life Sciences and Biomedicine* field between 1990 and 2020. Wos has been widely adapted to management studies (*Li, Rollins & Yan, 2018*; *Li, Chen & Wang, 2021*). In our work, we crawl the web and create the

**Table 1  Number of documents per time period after pre-processing for three datasets.**

| Periods | WoS | Periods | ACL | Periods | Enron |
|---|---|---|---|---|---|
| 1990–1994 | 6,391 | 1980–1984 | 551 | 1998-01–1998-11 | 68 |
| 1995–1999 | 12,094 | 1985–1989 | 919 | 1998-12–1999-04 | 461 |
| 2000–2004 | 19,020 | 1990–1994 | 1,903 | 1999-05–1999-09 | 4,259 |
| 2005–2009 | 29,347 | 1995–1999 | 2,458 | 1999-10–2000-02 | 18,401 |
| 2010–2014 | 40,910 | 2000–2004 | 3,947 | 2000-03–2000-07 | 50,797 |
| 2015–2019 | 50,791 | 2005–2009 | 7,410 | 2000-08–2000-12 | 115,777 |
| 2020–2020 | 12,946 | 2010–2014 | 11,977 | 2001-01–2001-05 | 136,026 |
|  |  | 2015–2019 | 18,822 | 2001-06–2001-10 | 77,927 |
|  |  | 2020–2022 | 15,212 | 2001-11–2001-12 | 36,103 |
| Total | 171,499 | Total | 63,199 | Total | 439,819 |

dataset till 2020, making sure the *corpus* is relatively up-to-date. We use abstracts together with the title as our *corpus*.

### ACL

The ACL anthology dataset (*Rohatgi, 2022*) contains publications in the domain of NLP and Computational Linguistics. The publications are conference articles, journal articles and workshop articles spanning from 1980 to 2022. The dataset contains 80,013 documents originally and 63,199 documents after pre-processing. Similarly, we use abstracts and the title of each document.

### Enron

The Enron email dataset contains email conversations within the energy company Enron Corporation from 1998 to 2001. The content of the dataset covers topics from business practices to organizational communication. Unlike scientific publications, we use email bodies and email subjects as our *corpus*. Note that some documents might contain more than thousands of tokens. As mentioned in Figure, we remove documents that are longer than 700 tokens for computational reasons, especially for BERTopic.

The two academic publication datasets, WoS and ACL tend to be more formal and have greater structure due to the peer-reviewed publication processes. However, the Enron email dataset contains more casual-style texts, including misspellings, informal language usage, *etc*. Additionally, the length of abstracts is often between 150–250 words, while the length of documents in emails can vary drastically.

We adopt different pre-trained sentence-bert models due to varying domains. We apply biomedical sentence-bert for the WoS dataset and the distilled-sentence-roberta model for the ACL and Enron datasets.

To sum up, in our experiments, we apply CoWords, LDA, and BERTopic to each dataset and extract topics over time. We then perform topic matching between models using top-word overlap (TWOR) and KL divergence to align equivalent topics across different methods. Once topics are matched, we analyze their prevalence trends to assess

**Table 2  Extracted topics with the highest score using two matching strategies.**

| | WoS—Overlap | WoS—KL divergence |
|---|---|---|
| CoWords | Liver, glucose, lipid, insulin, fat, diabetic, hepatic, obesity, fatty_acid, cholesterol | Absorption, mobile, bioelectromagnetic, guideline, electromagnetic, wileyliss, mobile_phone, antenna, phone, mhz |
| LDA | Evaluate, assess, estimate, glucose, consistent, plasma, obesity, estimation, differential, trial | Model, spatial, article, focus, derive, finally, account, mathematical, temporal, input |
| BERTopic | Diabetic, insulin, glucose, rat, diabetes, mouse, insulin_resistance, adipocyte, islet, hepatic | Cardiac, muscle, cardiomyocyte, heart, calcium, mouse, myocardial, cell, channel, skeletal_muscle |

| | ACL—Overlap | ACL—KL divergence |
|---|---|---|
| CoWords | Error, correct, grammatical, correction, spelling, spell, gec, error_correction, spelling_correction, misspell | Approach, base, set, compare, technique, accuracy, perform, apply, label, combine |
| LDA | Detection, error, detect, translate, correct, correction, check, sensitive, spelling, loglinear | Method, approach, propose, evaluate, score, focus, exist, baseline, outperform, metric |
| BERTopic | Error, grammatical, correction, gec, learner, error_correction, spelling_correction, spelling, detect, chinese | Dialogue, speech, dialog, speak, conversation, response, conversational, agent, recognition, speaker |

| | Enron—Overlap | Enron—KL divergence |
|---|---|---|
| CoWords | Time, deal, gas, market, power, service, price, day, energy, company | Ect, subject, message, forward, original, fax, hou, tomorrow, lon, confirmation |
| LDA | Price, offer, risk, lock, sfodenver, rls_tariff, vjw, faithbase, obliterate, farreache | Message, original, delete, civil_libertie, scarff, clintonappointe, groyer, faithbase, obliterate, bridget_maronge |
| BERTopic | Ect, email, subject, message, service, market, receive, gas, business, contact | Subject, ect, message, forward, email, agreement, attach, meeting, market, issue |

how each model detects topic emergence. Finally, we evaluate the models using both qualitative trend analysis and a quantitative emergence detection metric, which measures the agreement between local models trained on time-segmented data and a global model trained on the full dataset. This setup allows us to systematically compare the ability of different topic models to detect emerging topics across domains. We present the results of these experiments and discuss them in the following sections.

## RESULTS

This section presents the results of applying all three topic models (CoWords, LDA, and BERTopic) to each dataset. We first compare the TWOR matching and KL divergence qualitatively by examining the top-matched topics manually. We then present a quantitative evaluation using our proposed metric to measure the models' ability to detect emerging topics.

### Qualitative analysis

To compare TWOR and KL divergence, we examine their top-matched topics across models and datasets. Table 2 presents the topics with the highest matching scores for using each topic model, applying two matching strategies.

Our analysis shows that TWOR consistently retrieves semantically coherent topics, whereas KL divergence often prioritizes temporal trends over direct topic alignment. For

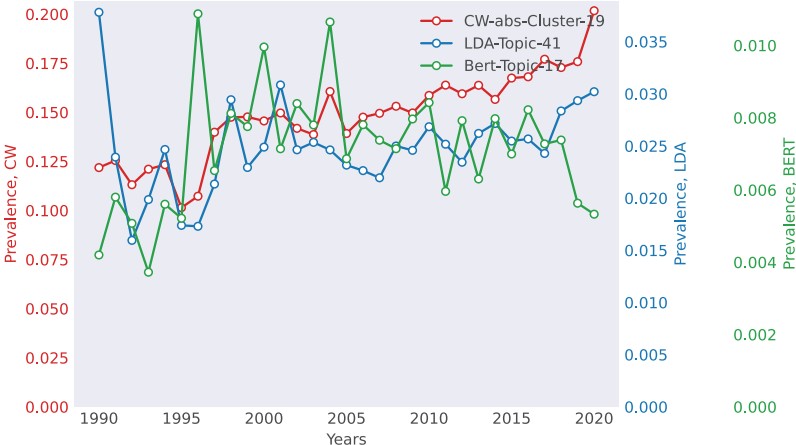

**Figure 3 Selected match for WoS. The top 10 words for each method are as follows.** CoWords: {infection, mortality, transmission, virus, spread, infect, vector, incidence, viral, epidemic}, LDA: {mouse, infection, antioxidant, virus, observed, stimulation, respond, protection, viral, infect}, BERTopic: {epidemic, infection, virus, viral, vaccination, model, transmission, vaccine, infectious, infect}.

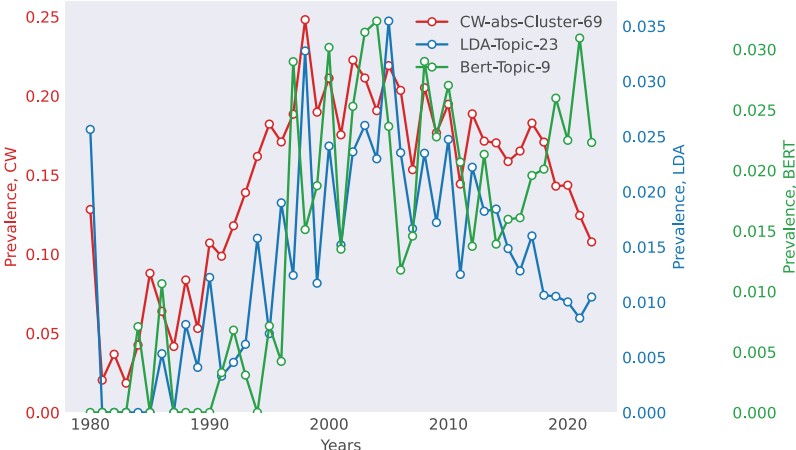

**Figure 4 Selected match for ACL. The top 10 words for each method are as follows.** CoWords: {segmentation, tag, Chinese, character, segment, boundary, tagging, tagger, wordlevel, partofspeech_tagged}, LDA: {accuracy, segmentation, character, segment, rich, morphology, morpheme, unsupervise, convention, lefttoright}, BERTopic: {segmentation, chinese, word, character, model, tagging, partofspeech_tagged, ngram, language, accuracy}.

example, in the WoS dataset, TWOR matches a topic centered on Diabetes (*e.g.*, "insulin," "glucose," "obesity"), while KL divergence instead retrieves bioelectromagnetic topics, likely due to similar growth patterns rather than shared semantics. A similar pattern emerges in the ACL dataset, where TWOR retrieves Grammatical Error Correction, while KL divergence retrieves broader methodological terms (*e.g.*, "baseline," "metric"). These results highlight that TWOR is better suited for direct topic alignment, while KL divergence may be more useful for finding temporally related but semantically distinct topics.

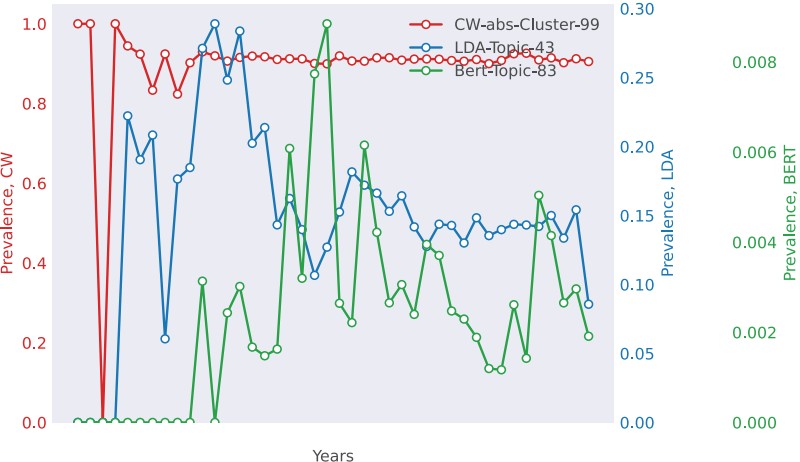

**Figure 5 Selected match for Enron.** The top 10 words for each method are as follows. CoWords: {email, agreement, question, receive, meeting, schedule, request, contact, file, list}, LDA: {agreement, contract, review, document, draft, bind, title, attorney, signature, wrong}, BERTopic: {abb_transformer, abb, existence, agreement, override, transformer, signature, initial, word, option}.

### Case study: topic evolution across domains

In this section, we present a case study to analyze how different models track emerging topics over time across the three datasets. We analyze the topics based on their extracted top words and their corresponding revolution.

**Case 1: Biomedical Research (WoS)—The Rise of Epidemiology** The WoS dataset captures the emergence of epidemiology-related topics, as shown in Fig. 3. TWOR-aligned topics (*e.g.*, "insulin," "glucose," "obesity") reflect a clear trend in diabetes research, with increasing prevalence over time. This aligns with global public health concerns and real-world epidemics like Ebola and SARS, which drove research interest in epidemiology. In contrast, KL divergence retrieves a bioelectromagnetics-related topic, demonstrating its tendency to group topics with similar temporal patterns rather than semantic alignment.

**Case 2: Computational Linguistics (ACL)—Advances in Word Segmentation** The ACL dataset provides insight into the development of Natural Language Processing (NLP) over time (Fig. 4). The matched topic reflects Grammatical Error Correction and Word Segmentation, showing a surge in research interest in the late 1990s and early 2000s, aligning with the rise of statistical NLP approaches. This suggests that topic models effectively track research trends, reinforcing the usefulness of retrospective analysis in scientific trend forecasting.

**Case 3: Corporate Communication (Enron)—Variability in Business Agreements** The Enron dataset presents a unique challenge due to its unstructured, informal text (Fig. 5). The matched topic centers around business agreements, but topic volatility differs across models. LDA and CoWords extract broader business-related terms, whereas BERTopic produces more volatile topic representations, possibly due to its reliance on sentence embeddings, which capture finer-grained variations in contract-related discussions.

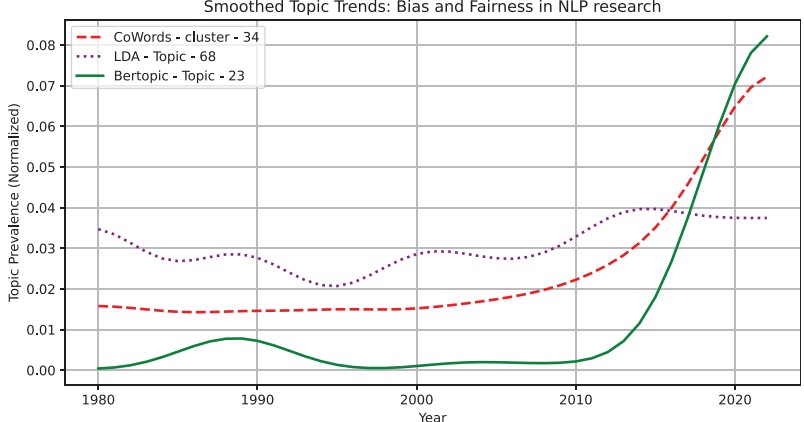

**Figure 6 Smoothed trends for the topic: Bias and Fairness in NLP research.** CoWords: {bias, gender, mitigate, age, demographic, biased, debiase, fairness, female, male}, LDA: {gender, mitigate, transe, people, production, rapidly, expectation, game, progress, produce}, BERTopic: {gender, pronoun, bias, pronoun_resolution, stereotype, female, debiase, language, gendere, stereotypical}.

**Case 4: A Emerging topic case—Bias and Fairness in NLP research** We present a case study where our models detect an emerging topic related to Bias and Fairness in NLP research. This topic matched across models, demonstrates how different topic modeling approaches capture the emergence of new research directions over time. Figure 6 presents its smoothed prevalence trends. We observe that BERTopic detects weak signals first (before 2000) but does not show sustained growth until post-2010, suggesting early semantic awareness but delayed recognition of topic prevalence. CoWords and LDA capture sustained emergence earlier, with CoWords showing a gradual increase from the early 2000s, while LDA detects a structured rise between 1995 and 2005. CoWords excels in tracking early co-occurrence shifts, LDA tends to stabilize and capture steady trends, and BERTopic is more sensitive to semantic shifts once the topic is widely established. The detected trends pattern aligns with real-world topic adoption trends. Early discussions on NLP bias existed before 2000, but they were scattered and lacked formal structure, similar to how BERTopic detects weak signals first. Between 2000 and 2010, fairness in NLP gained academic traction, reflected in LDA's structured rise. Finally, post-2010, NLP bias became a mainstream issue, driven by ethical AI debates and policy discussions, aligning with BERTopic's stronger detection at this stage.

From the qualitative analysis we observe the potential characteristics of each topic model for topic emergence ability. However, they lack systematic comparisons and quantified measure for evaluating their performance for the task. We then perform the quantitative analysis using our proposed metric.

## Quantitative analysis

Next, we evaluate the models' ability to detect emerging topics using our proposed evaluation metric. Since CoWords model does not generate document-topic probability distribution, the evaluation metric is not applicable. Figure 7 presents the average F1 scores

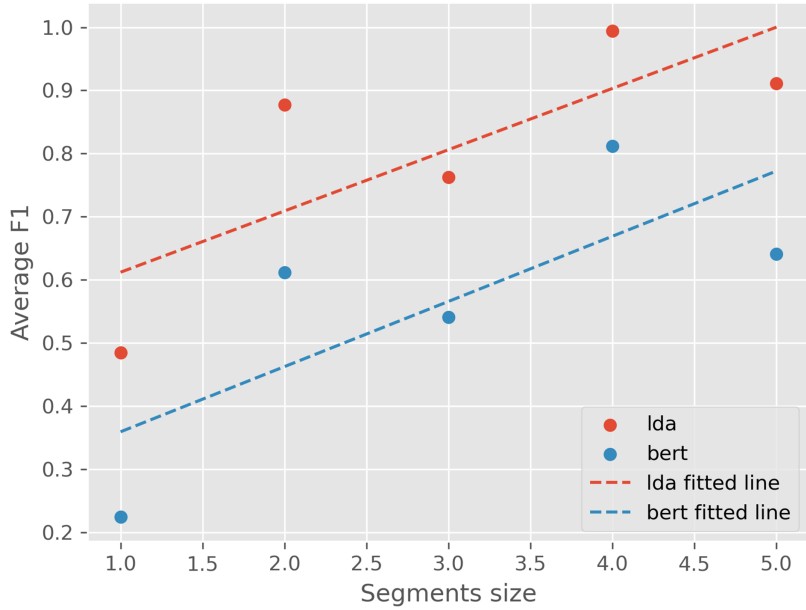

**Figure 7 Average F1 score for LDA and BERTopic given different segment sizes of the ACL dataset.** The overall average F1 score for LDA is 80.1%, and for BERTopic is 56.6%.

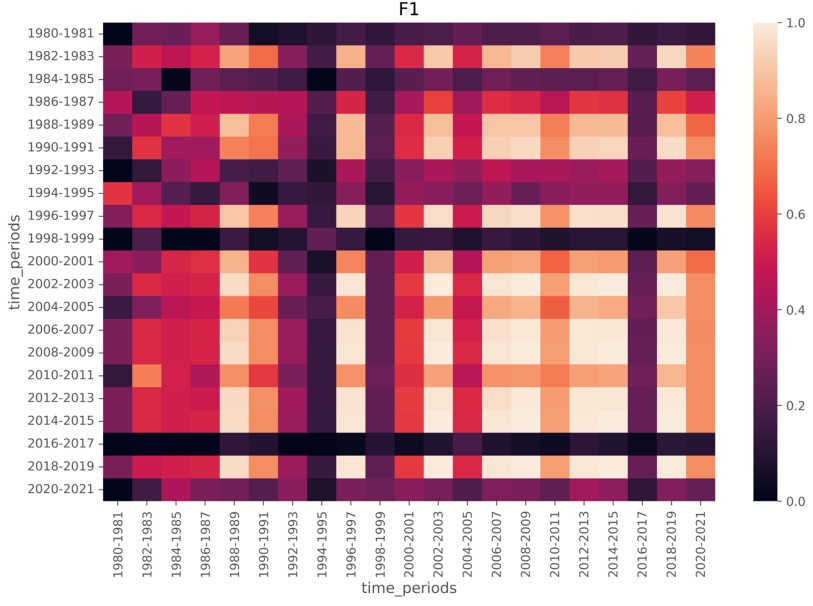

**Figure 8 Heatmap of F1 scores for LDA on segment size of 2.** For each cell, the model is trained on the training data from the row time periods (2-year span) and tested on the test data from the column time periods (2-year span).

between local models trained on different time segments and a global model trained on the full dataset. Results indicate that LDA achieves a higher average F1 score (80.1%) compared to BERTopic (56.6%), suggesting that LDA more consistently detects emerging

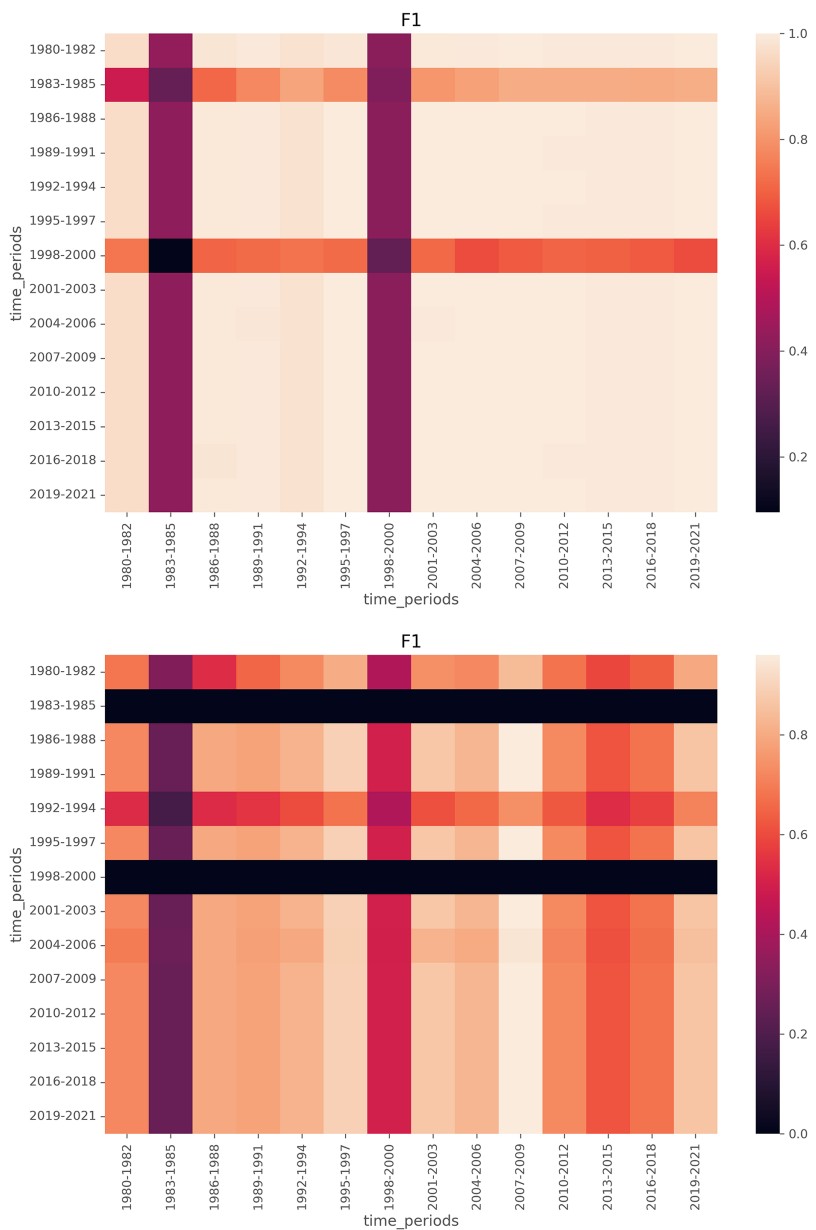

**Figure 9 Heatmaps of F1 scores for LDA (top) and BERTopic (bottom), with the segment size of 3, meaning local models are trained on training data from a 3-year span and tested on test data from a 3-year span.**

topics over time. Furthermore, results also show that F1 scores increase as segment size grows, suggesting that models trained on longer time spans capture more stable topic representations.

To further analyze how local models align with the global model, we visualize F1 score heatmaps in Figs. 8 and 9. We have a few key observations from the heatmaps. First, LDA achieves more stable F1 scores across time compared to BERTopic. Second, Periods of lower F1 scores (*e.g.*, 1980s, post-2015) indicate time spans where emerging topics were

harder to detect. Third, BERTopic's lower consistency suggests it is more sensitive to novel topics, while LDA generalizes better even with limited training data.

## DISCUSSION

Our findings reveal several key insights regarding matching strategies, dataset differences, model performance, and topic emergence detection effectiveness.

### Comparison of topic matching strategies

Our results show that TWOR consistently retrieves more semantically coherent topics across datasets, while KL divergence prioritizes topics with similar temporal trends rather than direct word overlap. This is evident in the WoS dataset, where TWOR identifies a topic closely associated with Diabetes, whereas KL divergence retrieves bioelectromagnetic-related terms. A similar pattern is observed in the ACL dataset, where TWOR effectively retrieves topics on Grammatical Error Correction, while KL divergence results in more general methodological terms.

These findings suggest that TWOR is more reliable for direct topic alignment, especially for structured datasets like WoS and ACL, where well-defined topics are expected. However, KL divergence can still be useful for detecting "related but distinct" topics, reflecting indirect relationships in topic evolution (*Uban, Caragea & Dinu, 2021*). In datasets like Enron, where topics are less structured, KL divergence may capture latent associations between business-related discussions.

### Dataset difference in topic extraction

Among the datasets we evaluated, we observed that topic quality varies depending on the level of structure in the text. WoS and ACL datasets yield clearer, more interpretable topics, likely due to the structured nature of research articles, which focus on well-defined subjects. The Enron dataset produces broader, more ambiguous topics, reflecting the informal and unstructured nature of email communications, where multiple themes may coexist in a single document.

These findings highlight the importance of dataset structure when applying topic modeling methods. While structured texts allow for more precise topic modeling, unstructured datasets may require additional preprocessing or more advanced modeling techniques to improve topic coherence.

### Comparison of topic extraction across models

Our analysis also highlights differences in how CoWords, LDA, and BERTopic extract topics. First, CoWords and BERTopic tend to extract topics with more overlapping words, likely due to their clustering-based approach. Second, LDA generates broader topics, with more diverse top words, whereas BERTopic and CoWords extract more specific topics (*e.g.*, "spelling" and "spell" appearing in the same topic). Third, BERTopic's topics appear more volatile, possibly due to its reliance on contextualized sentence embeddings, which may lead to greater variability in topic representations over time.

These findings suggest that LDA provides a more generalizable representation of topics, while BERTopic and CoWords tend to extract more granular topics with repeated terms.

This may explain why LDA aligns more closely with global trends, whereas BERTopic exhibits more sensitivity to transient or highly specific topics.

## Model performance on topic emergence detection

Given our qualitative analysis, we observe that CoWords performs better at detecting topic emergence than LDA and BERTopic. However, due to its inherent limitation on generating document-topic distribution, the quantitative metric is not directly applicable. Therefore the limitation of this conclusion is that it is based on limited qualitative results, and might be limited to generalize to a larger scale. On the other hand, evaluated by our proposed quantitative metric, we have a few observations. First, LDA detects emerging topics earlier and more consistently than BERTopic, supporting prior findings that probabilistic topic models perform well even with smaller datasets. Second, BERTopic's performance improves with larger time spans, likely due to its reliance on sentence embeddings, which require more data to generalize effectively.

While CoWords appears to detect emerging topics earlier, its limitations must be considered. Unlike LDA and BERTopic, CoWords does not produce document-topic distributions, which prevents its evaluation using quantitative metrics. Another limitation is that CoWords relies purely on word co-occurrence, lacking the semantic depth of LDA's probabilistic modeling or BERTopic's transformer-based embeddings. As a result, topics extracted by CoWords may be more fragmented or overlap significantly, particularly in datasets with subtle conceptual distinctions. In contrast, LDA and BERTopic provide more structured and interpretable topic representations, albeit at the cost of slower emergence detection.

## Impact of disruptive periods on emerging topics detection

Notably, the lower F1 scores in the 1980s and post-2015 suggest that emerging topics were harder to detect in these periods. This aligns with historical trends: early NLP research was sparse before the 1990s, while post-2015 saw rapid shifts in deep learning techniques. Our heatmap analysis further indicates that local models struggle to detect emerging topics when disruptive changes occur, a finding consistent with prior work on language model generalization across time.

## Lack of comparative evaluation in prior work

Evaluating topic models is inherently challenging, as traditional metrics like coherence and perplexity measure topic quality but fail to capture a model's ability to detect emerging topics. Prior work on emergence detection, such as BERTrend (*Boutaleb, Picault & Grosjean, 2024*) and ATEM (*Rahimi et al., 2023*), introduces novel methods for emerging trends detection. However, these studies focus on single-model evaluations rather than comparative benchmarking. Most studies assume that a single model is sufficient, overlooking the differences in how various models detect emergence over time. Our study highlights that CoWords detects emergence earliest, LDA provides stable trends, and BERTopic captures nuanced contextual shifts but exhibits higher volatility. These comparative insights are crucial because existing methods often rely on external metadata

(*e.g.*, citations or manually labeled emerging topics) rather than evaluating topic models independently. By proposing a self-contained evaluation metric, we provide a more robust framework for assessing when and how different models detect topic emergence, contributing towards more reliable evaluation strategies.

# CONCLUSIONS

In this study, we systematically evaluated three topic modeling approaches, CoWords, LDA, and BERTopic, for their ability to detect emerging topics across structured (WoS, ACL) and unstructured (Enron) datasets. Our analysis focused on topic matching strategies, differences in topic extraction quality, and an independent evaluation metric for assessing topic emergence detection.

Comparing two topic matching strategies, our findings indicate that TWOR consistently produces more semantically coherent topic matches than KL divergence, particularly in structured datasets such as WoS and ACL, where well-defined topics facilitate more precise alignment. KL divergence, on the other hand, tends to retrieve topics with similar temporal trends rather than direct word overlap, making it useful for identifying related but distinct topics. These results highlight the importance of choosing an appropriate matching strategy depending on the dataset structure and the intended analysis goal.

When evaluating on different datasets, we also observed notable differences in topic extraction quality across datasets. WoS and ACL datasets produced more interpretable topics, while the Enron dataset resulted in broader and more ambiguous topic clusters. This difference suggests that the effectiveness of topic models is influenced by the structure of the underlying *corpus*, with more formal and domain-specific texts yielding clearer topic representations.

When evaluating model performance in detecting emerging topics, our proposed metric showed that LDA consistently outperformed BERTopic in terms of agreement between local and global models. LDA achieved a higher average F1 score across all segment sizes, indicating its robustness in detecting emerging topics using limited data from specific time periods. BERTopic exhibited greater sensitivity to novel topics but showed lower overall consistency, which may be attributed to its reliance on sentence embeddings that require larger datasets to generalize effectively.

## Limitation and future work

Despite these insights, our study has some limitations. First, we evaluated topic emergence in a retrospective setting, meaning that our approach does not predict future topics but rather identifies when topics gain prominence over time. Second, while we examined three widely used topic models, future work could explore additional topic models to determine whether they provide advantages in emergence detection. Finally, our analysis focused on specific domain datasets (scientific articles, emails), and extending the study to other domains, such as news articles or social media data, could provide further validation of our findings.

In future work, we aim to expand our study to incorporate additional topic modeling approaches from the three categories, such as algebraic topic model non-negative matrix factorization (NMF) (*da Kuang, Choo & Park, 2015*) instead of CoWords, probabilistic topic model hierarchical Dirichlet process (HDP) (*Teh et al., 2004*) instead of LDA, as well as neural topic model Embedded Topic Model (ETM) (*Dieng, Ruiz & Blei, 2019*) instead of BERTopic. These models could provide deeper insights into semantic topic shifts and representation learning for emergence detection.

Additionally, we plan to extend our analysis to datasets from diverse domains, including finance, where emerging topics can influence market trends, and news or online communication, which often exhibit rapid topic evolution and linguistic variability. Exploring different textual styles, such as formal reports *vs* social media discussions, will help assess the adaptability of topic models across different communication formats.

### Funding
This work is funded by the Dutch Research Council (NWO) through grant MVI.19.032. The funders had no role in study design, data collection and analysis, decision to publish, or preparation of the manuscript.

### Grant Disclosures
The following grant information was disclosed by the authors:
Dutch Research Council (NWO): MVI.19.032.

### Competing Interests
The authors declare that they have no competing interests.

### Author Contributions
- Xue Li conceived and designed the experiments, performed the experiments, analyzed the data, performed the computation work, prepared figures and/or tables, authored or reviewed drafts of the article, and approved the final draft.
- Ciro D. Esposito conceived and designed the experiments, analyzed the data, authored or reviewed drafts of the article, and approved the final draft.
- Paul Groth conceived and designed the experiments, authored or reviewed drafts of the article, and approved the final draft.
- Jonathan Sitruk conceived and designed the experiments, authored or reviewed drafts of the article, and approved the final draft.
- Balazs Szatmari conceived and designed the experiments, authored or reviewed drafts of the article, and approved the final draft.
- Nachoem Wijnberg conceived and designed the experiments, authored or reviewed drafts of the article, and approved the final draft.

## Data Availability

Data and code are available at https://zenodo.org/records/14503316.

## Supplemental Information

Supplemental information for this article can be found online at http://dx.doi.org/10.7717/peerj-cs.2875#supplemental-information.

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
