# Peer review of "Evaluation of unsupervised static topic models’ emergence detection ability"

_PeerJ Computer Science, doi:10.7717/peerj-cs.2875_

## Round 0.1 · original submission · Major Revisions

Your work is interesting and valuable. However, there are some issues as reviewers mentioned. Please revise the work carefully according to the comments.

Reviewer 1 ·

Basic reporting

The whole report is given in the "Additional comments" section.

Experimental design

The whole report is given in the "Additional comments" section.

Validity of the findings

The whole report is given in the "Additional comments" section.

Additional comments

Evaluation of unsupervised static topic models' emergence detection ability

This study focuses on addressing the gap by exploring three key static, unsupervised topic extraction techniques: CoWords clustering, Latent Dirichlet Allocation (LDA), and Bertopic, with a specific focus on their effectiveness in identifying topic emergence. The authors conducted a thorough qualitative analysis of the topics extracted and their emergence across three datasets: biomedical publications from the Web of Science, publications from the ACL anthology, and the Enron email dataset. Furthermore, they introduce a quantitative measure to assess LDA and Bertopic independently regarding their capacity to identify topic emergence without relying on ground truth. This comparative analysis reveals each method's relative strengths and weaknesses, offering valuable insights into their applicability and performance across various textual contexts.

The study looked technical and was well prepared. However, It has some significant concerns, as listed below:

1- The abstract must contain the main numerical results.

1- The aim of the study, the gap in the literature, the main focus of the study, and the challenges of the previous works must be given in the introduction section. The novelty of the study is not explicit.

The authors summarized the study's key contributions with three bullet points at the end of the introduction. However, none is an exact contribution. They are only methodological and material-based explanations. It must be explained more.

2—The literature survey must be extended to include the most recent studies.
The authors mentioned very few studies in the literature review section. It is far from taking a photo of literature.

3—The proposed model was not explained clearly. The explanations in the methods section are bookie information. The proposed model's details should be more detailed and technical.

4- In Figure 1, the proposed model steps must be detailed.

5—The authors state that the emergence detection measurement (EDM) is related to the growth rate within any given time window, as shown in Eq. 5. This approach should be compared with known EDMs to show the effectiveness of the proposed approach.

6- The results and discussion section should be given as separate sections.

7- In the discussion section, the authors must discuss the results of previous works.

8- The merits and demerits of the study over similar studies should be given in the discussion section.

9— The study must be checked grammatically.

10- The conclusion section should be extended.

Cite this review as

Reviewer 2 ·

Basic reporting

I have provided my comments regarding the study in the "Additional Comments" section, which include my observations on areas where I found missing details or opportunities for improvement. Please update the manuscript by addressing the revisions and comments provided, as they will strengthen the overall quality of the work.

Experimental design

I have provided my comments regarding the study in the "Additional Comments" section, which include my observations on areas where I found missing details or opportunities for improvement. Please update the manuscript by addressing the revisions and comments provided, as they will strengthen the overall quality of the work.

Validity of the findings

I have provided my comments regarding the study in the "Additional Comments" section, which include my observations on areas where I found missing details or opportunities for improvement. Please update the manuscript by addressing the revisions and comments provided, as they will strengthen the overall quality of the work.

Additional comments

Dear Authors
Your study is promising and provides valuable insights into topic emergence. However, I noticed some areas where there are missing details, and I have outlined these as separate comments. With the suggested revisions, I believe the paper will not only improve in clarity and depth but will also make an important contribution to the field. Additionally, it is important that any comments that could not be addressed are clearly stated in the limitations section of the paper. I am confident that incorporating these points will enhance the rigor of your work, and I look forward to seeing the updated version.


Comments:

Method Selection: The methods used in the study are well-chosen, as comparing CoWords clustering, LDA, and Bertopic provides an important approach to understanding topic emergence in different types of texts. However, including additional methods, particularly other deep learning-based models, could have expanded the scope of the study.

Datasets: The three different datasets used in the study (Web of Science, ACL Anthology, and Enron) are quite diverse, enhancing the generalizability of the analysis in different contexts. However, a more detailed explanation of the content and features of each dataset would increase the transparency of the study.

Comparison of Methods: The combination of qualitative and quantitative analyses used to compare methods is effective. However, providing deeper examples of qualitative analysis could help readers understand more clearly how the topics were identified.

Generalizability of Results: The study's findings are limited to only three datasets, which hinders general conclusions. Suggestions for future work could involve expanding the study to include additional datasets to test the methods' performance in a broader context.

Quantitative Metrics: The success of LDA and Bertopic in detecting topics has been evaluated using a quantitative measure, which provides an objective comparison. However, a more detailed explanation of these quantitative metrics could provide further insight into why the methods succeeded or failed.

Clarity of Language and Expression: The overall language of the paper is clear and understandable. However, some technical terms and concepts could be explained in simpler terms or with more detail for readers unfamiliar with the topic.

CoWords Clustering Method: The findings showing CoWords clustering as superior for topic detection are interesting. However, the limitations and potential drawbacks of this method should be addressed in more detail.

Comparison of Scientific Publications and Email Data: The comparison between scientific papers and email data clearly demonstrates how different contexts affect topic emergence. However, it would be useful to emphasize that adding a wider variety of datasets could enrich the scope of the study and guide future research.

Literature Review: The literature review is well-connected with existing research. However, it would be beneficial to examine more recent studies and compare them with the approaches used in this paper.

Suggestions for Future Work: Future studies could benefit from incorporating larger datasets and a more diverse range of methods. Additionally, research into the impact of deep learning-based approaches for topic detection could be valuable.

Preprocessing Steps: More detailed information on the preprocessing steps would help clarify which data cleaning and text preprocessing techniques were used. For example, explanations of whether stop-word removal, lemmatization, or tokenization were performed would be useful. Discussing the specific preprocessing methods used for each dataset would further improve transparency.

Choice of Three Methods: The selection of CoWords, LDA, and Bertopic is a good approach for ensuring diversity in topic modeling processes. However, the study would benefit from explaining why other topic modeling techniques (such as HDP or NMF) were not chosen. The reasons behind the selection of these three methods, including a comparison of linear and nonlinear models, should be discussed more clearly.

Emergence of New Topics in Topic Clusters: Illustrating the emergence of new topics within a topic cluster would help readers better understand the process. This type of visualization would provide insight into the dynamic nature of topic modeling and how it evolves over time. A concrete example of the emergence of a new topic could be included.

Contribution and Importance of Results: The results of the study contribute to the literature and fill an important gap in topic modeling research. However, comparisons with previous studies should be made more explicitly, highlighting how this work addresses gaps or shortcomings of previous approaches. For example, if no similar comparisons have been made before, or if different datasets were not tested with the methods, it would emphasize the importance of this research.

Cite this review as

---

## Round 0.2 · accepted · Accept

Thanks a lot to the authors for their effort to improve the work. This version can be accepted currently. Congrats!

Reviewer 1 ·

Basic reporting

This version of the article has improved the technical quality of the paper. The authors handled my issues, so the study should be accepted.

Experimental design

This version of the article has improved the technical quality of the paper. The authors handled my issues, so the study should be accepted.

Validity of the findings

This version of the article has improved the technical quality of the paper. The authors handled my issues, so the study should be accepted.

Additional comments

This version of the article has improved the technical quality of the paper. The authors handled my issues, so the study should be accepted.

Cite this review as

Reviewer 2 ·

Basic reporting

The authors have thoroughly addressed the previously suggested comments regarding basic reporting. The application of topic modeling across three distinct datasets has notably strengthened the clarity and comprehensiveness of the study.

Experimental design

Following the recent revisions, the experimental design and methodology now effectively align with the study’s objectives and overall structure. The comparative analysis of CoWords clustering, LDA, and BERTopic demonstrates a well-constructed and rigorous experimental framework.

Validity of the findings

The validation, comparison, and citation of findings are appropriate and sufficient for the scope of the study. Supporting the results with relevant references and offering directions for future research have contributed to the maturity and depth of the work.

Additional comments

I have reviewed your manuscript entitled "Evaluation of Unsupervised Static Topic Models’ Emergence Detection Ability" in two rounds and provided detailed feedback during the process.
In the initial round, I suggested minor revisions, as I believed the study had the potential to contribute meaningfully to the field. I appreciate that the authors have addressed these suggestions with care, improving the clarity and coherence of the manuscript.
The current version of the paper reflects a well-developed study with valuable scientific insights.

In this context, I find the article suitable for publication in its current form.

Cite this review as